

# An acoustic-based method for locating maternity colonies of rare woodland bats

Kieran D. O'Malley[1], Henry Schofield[2], Patrick G.R. Wright[2], Daniel Hargreaves[2], Tom Kitching[2], Marina Bollo Palacios[2] and Fiona Mathews[1]

[1] School of Life Sciences, University of Sussex, Brighton, East Sussex, United Kingdom
[2] Vincent Wildlife Trust, Ledbury, Herefordshire, United Kingdom

## ABSTRACT

Locating colonies of rare bats can be a time consuming process, as it is often difficult to know where to focus survey effort. However, identifying peaks of bat activity via acoustic monitoring may provide insights into whether a colony is locally present, and help screen out sites with low potential. Using a triage approach, we developed a survey methodology for locating colonies of the woodland-specialist barbastelle bat (*Barbastella barbastellus*). We investigated whether woodland occupancy by a colony could be predicted by acoustic data, and assessed the influence of survey effort (number of acoustic detectors deployed) on detectability. The methodology was then trialled in citizen science surveys of 77 woodlands, with follow-up radio-tracking surveys by specialists being used to confirm presence or absence. Using Receiver Operating Characteristic (ROC) curve analysis, we found that a threshold of four barbastelle passes recorded by at least one detector within one hour of sunset optimised the balance between the true- and false-positive rates. Subsequently, we found that a minimum survey effort of one detector per 6.25 hectares of woodland was needed to ensure a colony would be detected using this threshold, based on a survey sensitivity of 90%. Radio-tracking surveys in a subset of the woodlands, identified as having a high probability of being occupied by a colony based on acoustic monitoring, confirmed the presence of five previously unknown barbastelle maternity colonies. These results demonstrate that a triage system, in which high probability woodland sites are identified based on acoustic survey data, can be used to prioritise sites for future specialist surveys and conservation action.

## INTRODUCTION

The ability to locate rare species is a central challenge for ecologists aiming to confirm the presence and population size of threatened or declining wildlife, or to assess their conservation status (*Venette, Moon & Hutchison, 2002*; *Berec et al., 2015*). Detectability is influenced by the trade-off between survey precision, geographical coverage, and survey effort, and is a key decision for many schemes studying rare species.

The barbastelle (*Barbastella barbastellus*; Schreber, 1774) is a rare, medium-sized vespertilionid bat classified as 'Near Threatened' globally by the International Union for

Corresponding author
Fiona Mathews,
f.mathews@sussex.ac.uk

the Conservation of Nature (*Piraccini, 2016*), and 'Vulnerable' in Great Britain (*Mathews & Harrower, 2020*). Across Europe, the loss of ancient woodlands and reduction in insect prey has led to historic declines (*Carr et al., 2020*), though accurate estimates of their current population size are unknown (*Piraccini, 2016*). Its narrow ecological niche and preference for roosting under loose bark and within crevices of old or dead tree trunks means populations are highly dependent on mature broadleaved woodlands (*Sierro & Arlettaz, 1997*; *Russo et al., 2004*). In southwest England, barbastelles have been shown to select standing dead oak (*Quercus* spp.), as well as trees with a more open canopy and those in close proximity to water (*Carr et al., 2018*). Whilst our understanding of the distribution and use of woodlands by barbastelle colonies has improved, knowledge gaps remain owing to the difficulties associated with their detection.

Ecological assessments are commonly undertaken for bats in Europe as they are given high legal protection (*e.g.*, European Union Habitats Directive [92/43/EEC]). Whilst good practice guidelines for undertaking bat surveys are available (*Battersby, 2010*; *Collins, 2016*), challenges remain since bats are small, nocturnal, volant, and often difficult to identify to species (*Barlow et al., 2015*). Using a combination of survey techniques, for example acoustic surveys, visual inspections, or trapping, is regularly proposed as a means to minimise sample bias (*Milne et al., 2004*; *Flaquer, Torre & Arrizabalaga, 2007*; *MacSwiney, Clarke & Racey, 2008*; *Lintott et al., 2014*; *Braun de Torrez, Ober & McCleery, 2016*), however this is often not logistically or financially feasible. Key data gaps still remain that hinder the evaluation of conservation status *e.g.*, Red List assessments, and determining the distribution of roosts is critical for developing effective conservation strategies (*Russo et al., 2004*; *Wiederholt et al., 2015*). The process of locating colonies is currently heavily reliant on the capture and radio-tracking of individual bats, making it an expensive and time-consuming process (*Vonhof & Barclay, 1996*). In contrast, passive acoustic monitoring has rarely been explored as a tool to locate bat roosts.

During the emergence period, bats generally stay within close proximity to their roost, before dispersing to foraging areas. *Braun de Torrez, Ober & McCleery (2016)* demonstrated that hot-spots of bat activity close to emergence times could help locate roosts of the endangered Florida bonneted bat *Eumops floridanus*, whilst simultaneously minimising costs. Identifying peaks of bat activity *via* acoustic monitoring may therefore provide insights into whether a colony is locally present (*Hill et al., 2015*). However, this has not been attempted in a systematic way for bats anywhere in Europe. This project has therefore developed a methodology based on acoustic monitoring, validated by radio-tracking, to identify roost locations.

To date, the design of acoustic surveys aimed at determining site-level species presence has largely focused on temporal variation generated by factors such as weather conditions (*Hayes, 2000*; *Flaquer, Torre & Arrizabalaga, 2007*; *Scanlon & Petit, 2008*; *Fischer et al., 2009*; *Meyer, 2015*), which can be addressed by increasing survey duration. Spatial variability, generated by the clustering of activity in certain regions, such as near to roosts or particular habitats, is more rarely considered (*Vaughan, Jones & Harris, 1997*; *Rodhouse, Vierling & Irvine, 2011*).
It is important to recognise that optimizing a survey methodology depends on the specific objectives. Of 460 studies that used passive acoustic monitoring in terrestrial environments, over 70% deployed a single recorder per site (*Sugai et al., 2019*). This approach could potentially be suitable for comparing species richness and relative activity between sites, provided detectors are deployed for sufficiently long, and temporal variation is taken into account. However, to gain a better understanding of the subtle differences between sites and for the identification of areas of concentrated activity, survey effort should also focus on accounting for spatial heterogeneity within sites. In a study of five woodland sites, *Fischer et al. (2009)* demonstrated that whilst night-to-night variation accounted for a larger portion of variability in bat activity (20%), spatial heterogeneity within a site (10%) still played a significant role. Thus, achieving a balanced approach to survey design requires careful consideration of both temporal and spatial factors.

Recent advancement in acoustic technology and digital signal processing has brought with it new opportunities for bat survey data to be collected by non-specialists. Provided adequate volunteer training and clear study designs (*e.g.*, sampling variability and frequency, use of stratified random sampling, standardised protocols), public contributions to acoustic monitoring may enable the survey of bats across large geographic scales (*Newson, Evans & Gillings, 2015*; *Brown & Williams, 2019*; *Armstrong et al., 2020*; *Lundberg et al., 2021*). Citizen science projects, such as the Irish Bat Monitoring Programme and The National Bat Monitoring Programme in the UK, have already demonstrated that bat activity trends can be modelled using data collected by trained volunteers (*Barlow et al., 2015*; *Aughney, Roche & Langton, 2022*).

In this project we aimed to quantify (i) an optimal activity cut-off threshold used to diagnose colony presence in woodlands, (ii) the survey effort (detector density and duration of monitoring) required to identify a woodland as containing a colony, and (iii) an optimal activity cut-off threshold used to identify woodlands with a high probability of capture success from trapping surveys. We address these aims in order to produce a citizen science survey methodology that could be widely applied to woodlands across Europe.

# MATERIALS & METHODS

## Phase 1: Acoustic methodology development

We collected acoustic data between May and September 2019 in 13 ancient semi-natural and replanted woodlands in the counties of West Sussex, Wiltshire, and Herefordshire in the UK. We based site selection on (i) previous records of known barbastelle colonies ($n = 6$) or (ii) the presence of ancient woodland where colonies were believed to be absent ($n = 7$). Sites were categorised as having barbastelle colonies or not on the basis of historical evidence from radio-tracking studies, trapping and/or bat box checks completed by third parties. We considered historical evidence as any surveys conducted within the five years preceding our study period (2014–2019). In addition, we analysed the temporal activity patterns in the acoustic data that was subsequently collected: we expected sites with colonies present to show clear peaks in activity close to the emergence time of barbastelles, as well as near dawn. In cases where we were unable to discern whether colonies were present or
absent with a reasonable level of confidence, we categorised the sites as 'unknown' and used them to test our models. The woodlands selected ranged in size from 14 to 122 ha, with elevations between 15 m (50°57′02″N, 0°36′26″W—Burton & Chingford Ponds Nature Reserve) and 203 m (52°4′43″N, −2°58′8″W—Moccas Park National Nature Reserve). Climatic conditions in the region are mild and wet with mean summer (May–September) temperatures between 14.4 and 15.2 °C and mean precipitation between 275 and 284 mm (*Hollis et al., 2019*).

We collected acoustic data at 146 survey points using full-spectrum passive Song Meter SM2BAT static detectors with SMX-US or SMX-U1 omni-directional microphones (Wildlife Acoustics, Maynard, MA, USA; see Article S1 for full detector settings). We spaced the detectors approximately 150 m apart along trails and rides, features known to be favoured by barbastelles (*Greenaway, 2004*), deploying 5–15 detectors per woodland depending on size and accessibility of the site. A strict grid was not used because some areas were inaccessible; and others suggest that the type of spatial arrangement used has minimum impact on detectability of rare species (*Berec et al., 2015*). To maximise detectability, we placed microphones at a height of approximately 1.5 m above the ground and orientated them horizontally and away from vegetation (*Weller & Zabel, 2002*). We checked microphone sensitivity prior to the start of the survey period, as well as approximately halfway through, using an ultrasonic testing device (Ultrasonic Calibrator; Wildlife Acoustics) with a 40 kHz pulse. Data obtained from nearby weather stations indicated that conditions complied with best practice guidelines for bat surveys in the UK (sunset temperature $\geq 10$ °C, no rain or strong wind; *Collins, 2016*) on all study nights (see Article S2 for recorded conditions).

Bat calls were distinguished from background noise using SonoBat software (version 4.5.0, SonoBat, Arcata, CA, USA). All remaining files were then subjected to manual checking, with reference to parameters provided in *Russ (2012)*. We considered a file to be equivalent to a single bat pass (traditionally defined as a call or series of calls, separated by no more than a one second time gap; *Fenton, 1970*), as preliminary analysis indicated that multiple passes of barbastelles occurred in only 2.7% of files. As barbastelles recorded within the first hour after sunset fall within the species-specific emergence time range (*Russ, 2012*), passes close to dusk may also be indicative of a nearby roost. Therefore, we conducted analysis on activity recorded within the first hour after sunset. Whilst high activity close to dawn may be indicative of a nearby roost, we did not analyse this time period as previous work found return times to be highly variable among barbastelle individuals ($\bar{X} = 194 \pm 59.1$ min before sunrise; *Zeale, Davidson-Watts & Jones, 2012*), whereas emergence times were much more consistent.

## Woodland coverage maps

A given unit increase in detector number has a smaller impact on detector density as woodland size increases. There is also likely to be more heterogeneity of habitat in larger parcels compared with smaller ones. As such, there is a need to standardise surveys to account for the spatial variation in activity, as well as the size of a site. We therefore estimated detector density by constructing smoothed woodland coverage maps by fitting concave

hulls (alpha hulls) to detector locations in QGIS (version 3.6.3-Noosa; *QGIS Development Team, 2020*). The alpha hull is an algorithmic method for assigning a boundary around a discrete set of points. They are constructed by creating a Delaunay triangulation of all points within the sample and subsequently retaining only those vertices which are shorter in length than the chosen value of the parameter alpha (*Burgman & Fox, 2003*). Varying the value of alpha between zero (*i.e.,* a set of discrete points) and one (*i.e.,* a minimum convex polygon) generates different hull configurations, which will include increasingly isolated points as the value increases. In this study, we set the value of alpha to 0.7, representing the best balance between the inclusion of all points and the exclusion of areas without any detectors (for additional methodological details, see Fig. S1 and Article S3). We used alpha hulls rather than minimum convex polygons to minimise spatial bias and overestimation, which are particularly problematic for range and distribution estimates (*Burgman & Fox, 2003*).

## Phase 2: Application of methodology

We undertook acoustic bat surveys in four regions across England between May and September in 2021 (Fig. 1). Each region consisted of a 35 × 20 km area, in the counties of West Sussex, Wiltshire, Herefordshire, and the South Midlands (Northamptonshire, Buckinghamshire, Bedfordshire). The regions represented contrasting agricultural landscapes, and were all within the core geographical range of the species.

Within these regions, we selected study sites from broadleaved and mixed woodland in the National Forest Inventory. A stratified random sampling approach was used, using woodland patch size as the strata. Most woodland patches in England are small, and therefore simple random selection of study sites would have yielded a sample with few large woodlands, preventing any possible future investigation of the association between patch size and probability of occupancy. In addition, woodland size may be a constraint on barbastelle colony formation. This has been demonstrated for the Bechstein's bat (*Myotis bechsteinii*), a rare woodland bat with similar woodland structural requirements, which has been shown to need around 70 ha of good quality habitat to support colonies of 20–30 females in the Southern Upper Rhine region (*Steck & Brinkmann, 2013*). We therefore defined the minimum size category based on the smallest woodlands with barbastelle records, according to the most comprehensive available national assessment (*Mathews et al., 2018*). The remaining three categories were based on equal count quantiles of all deciduous woodland that had barbastelle records: (i) very small ≤ 1.04 ha, (ii) small > 1.04–18.48 ha, (iii) medium > 18.48–68.67 ha, and (iv) large > 68.67–1779.29 ha. We randomly selected five woodlands from each size category per region, giving a total of 80 sites. If we could not obtain landowner permission for a given woodland, the nearest accessible woodland that fell within the same size category was selected.

Volunteers were recruited from Spring 2021 to participate in the Barbastelle Volunteer Bat Survey (https://www.vwt.org.uk/research-all/monitoring-for-the-near-threatened-barbastelle/) and provided with pre-set acoustic detectors (Wildlife Acoustic SM2 and SM2+ detectors recording in full-spectrum at either 192 or 384 kHz, respectively). The number of units provided was dependent on the size of the woodland to ensure approximate
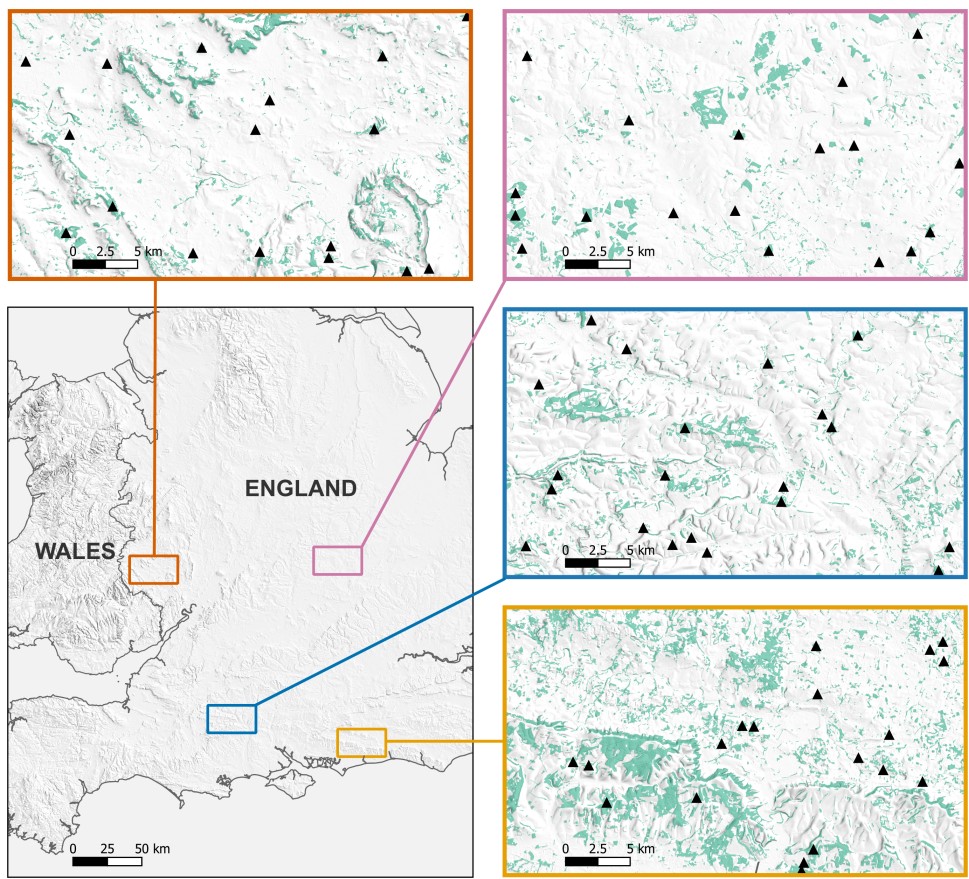

**Figure 1** **Location of the 77 woodlands acoustically surveyed for *B. barbastellus* across the four study regions.** Each surveyed woodland is represented by a black triangle. Green areas represent broadleaved or mixed woodland. Maps produced using QGIS version 3.6.3-Noosa (*QGIS Development Team, 2020*). Contains, or is based on, information supplied by: the Forestry Commission (©Crown copyright and database right 2021 Ordnance Survey (100021242)); MapTiler (©MapTiler ©OpenStreetMap contributors); Office for National Statistics (licensed under the Open Government Licence v.3.0. Contains OS data ©Crown copyright and database right 2023).

equal detector density across sites. Both the density of detectors used and the duration of the survey were determined based on the findings of the methodological development phase. Volunteers were instructed to place detectors along trails or rides (unless none were available) in the pre-selected woodlands. We provided a GPX file that included the recommended location of each detector, allowing volunteers to download and view the survey locations on a mobile phone, and asked them to maintain an approximately even distribution of detectors through the woodland if they had to deviate from the recommended locations. Volunteers recorded the grid reference and unique ID number of each detector on a recording form. For each survey, we used the same microphone placement, detector settings, acoustic analysis, and minimal weather conditions as adopted during the development phase. Once the survey was completed, volunteers returned all equipment back to one of the survey bases for analysis. We provided volunteers with

feedback in the form of a detailed survey report that summarised the bat species and barbastelle activity recorded at each detector.

## Phase 3: Validation by radio-tracking

We undertook trapping surveys in 2022 within a subsample of the woodlands acoustically surveyed by volunteers in 2021. Initially, we separated all woodlands into two categories (high and low probability of colony presence) based upon the optimal cut-off threshold identified in the development phase (see statistical analyses, below). Trapping was conducted in one randomly-selected low probability woodland in each of the four study regions. A further 13 high-probability woodlands were surveyed, with woodlands distributed approximately evenly between the study regions. Due to time availability within the season and logistical constraints, we prioritised surveying high-probability woodlands as our primary aim was to assess whether maternity colonies could be located using our methodology.

Nets were set at the detector location that recorded the highest number of barbastelle passes (defined as the highest number of barbastelle passes recorded within an hour of sunset, across all nights a detector was deployed). If there were no barbastelles passes at any detector within this time period (occurred only in the low-probability woodlands), then the location with the most barbastelle passes across the entire night was used. When we recorded no barbastelles within a woodland across the entire night, then the location was randomly selected from amongst the detector locations. If a location was deemed unsuitable (*i.e.,* due to accessibility and practicality) then we selected the next suitable location based on the criteria described above.

A combination of monofilament mist nets (total length = 45 m; Ecotone, Poland) and two harp traps (Faunatech Austbat, Victoria, Australia) was used at each site. A Sussex Autobat lure (Autobat Mk 2; Autobat, Sutton Coldfield, UK) broadcasting barbastelle echolocation and social calls, either in isolation or in sequences including other species, was used in conjunction with harp traps. Nets were opened for two hours from sunset and each location was surveyed for two nights. Adult female or juvenile barbastelles were fitted with lightweight radio transmitter PicoPip (Ag377, 0.29 g; LoTek, Ontario, Canada) or Holohil (LB-2X, 0.27 g; Holohil, Ontario, Canada) tags weighing < 5% of the bats weight. We clipped the fur of each bat and attached a tag between the scapulae using a flexible latex-based glue (either Ostomy Adhesive Salts Healthcare Ltd., Birmingham, UK, or Torbot Bonding Cement, Torbot Ltd, Cranston, RI, USA). We tagged a maximum of 2–3 individuals at each woodland site, and priority was given to parous females, followed by juvenile animals (identified by the lack of fused epiphyses), given the objective of identifying maternity roosts.

Tagged bats were tracked to roost trees the following day using Australis or Regal (Titley Electronics, New South Wales, Australia), Sika (Biotrack Ltd, Wareham, United Kingdom), or Biotracker (Lotek, Newmarket, Canada) VHF receivers with three-element Yagi antennas. We used an omni-directional antenna mounted on a vehicle to assist with tracking outside study woodlands. Possible occupied tree cavities were identified *via* visual inspection. Emergence surveys were then conducted, starting at sunset and lasting for 1.5 h

or until bats had finished emerging. Video recordings to permit assessment of the number of bats emerging were made using Canon XA20 and XA40 infrared cameras (Tokyo, Japan), mounted on tripods and positioned on either side of the tree, in conjunction with two infrared illuminators (LIR-IC88, 850 nm; IRLAB, Shenzhen, China; range = 180 m; beam angle of 40°).

## Statistical analysis

We conducted statistical analyses in RStudio using R version 3.6.1 (*R Core Team, 2019*) and produced graphical outputs using the *ggplot2* package (*Wickham, 2016*). The residuals of each model were checked for normality and heteroscedasticity using the 'testResiduals' function in the *DHARMa* package (*Hartig, 2022*).

For phase 1, we analysed the activity within the first hour after sunset in order to identify a cut-off threshold useful for distinguishing woodlands with and without barbastelle colonies. Where the full hour after sunset was not recorded completely, owing to occasional detector failure (approximately 5% of deployments—24 detector-nights in total), it was excluded. We took the maximum activity recorded across all the detectors within an hour of sunset to represent the best indication as to whether a given site was occupied by a colony. For each night, we selected the detector recording the highest level of activity for each site where the presence or absence of colonies was known (*i.e.,* excluding sites categorised as unknown).

We analysed the relationship between the pass rate and the presence/absence of a colony using a generalised linear model (GLM), specifying a binomial error structure with a logit link function. Colony presence was the response variable, bat passes within the first hour after sunset was the predictor, and each woodland was treated as a replicate. To determine the predictive power of the model and to calculate the optimal cut-off threshold for bat passes, we derived a receiver operating characteristic (ROC) curve using the package *pROC* (*Robin et al., 2011*). ROC statistics are informative in evaluating binary class decisions and cut-off parameterisation based on a models' true-positive (sensitivity) and false-positive rate (1 - specificity) at various threshold values. We used the 'coords' function in *pROC*, specifying ''best'' as an argument, to select the threshold corresponding to the best sum of sensitivity and specificity respectively.

To determine the survey effort (*i.e.,* detector density) required to identify the presence of colonies during phase 1, we constructed power curves using a random subsampling with replacement approach. For each site with a known or predicted colony, we calculated the density of detectors per smoothed alpha hull, then incrementally eliminated detectors randomly. Each time a detector was removed, the density was recalculated along with the maximum number of barbastelle passes that were recorded from the new subset. We compared this value with the optimal cut-off threshold derived from the ROC curve generated from the entire dataset, and categorised woodland as likely to contain a colony (value of 1) or not (value of 0). We estimated survey sensitivity (*i.e.,* probability of detecting a colony by at least one sample) as the proportion of 1,000 stochastic simulations in which at least one static detector had pass rates higher than the ROC-derived threshold.

We modelled the relationship between detector density and survey sensitivity using a generalised additive mixed model (GAMM), specifying a binomial error structure (*mgcv*

package; *Wood, 2011*). Detector density was specified as a predictor, fitted with penalised thin plate regression splines (knots = 3), and survey night was specified as a random effect. We used model predictions to determine the detector density required to maintain a specified minimum survey sensitivity. Following the precautionary principle, to minimise the possibility of concluding that no colony was present when in fact there was one (false-negative, Type II error), we adopted a stringent survey sensitivity of 0.9 (*Jones, 2013*; *Niver et al., 2014*; *Meyer, 2015*).

To determine the appropriate survey duration to be used in phase 2, we analysed the effect of duration on the prediction of colony presence. We conducted this analysis for two sites where we had complete acoustic data sets for five consecutive nights. For the full range of detector densities, we calculated the survey sensitivity for every possible combination of nights that could have been surveyed (1–5 nights). We modelled this relationship using a generalised additive model (GAM), specifying a binomial error structure (mgcv package; *Wood, 2011*). Survey sensitivity was used as our response variable, whilst detector density and number of survey nights (both fitted with penalised thin plate regression splines - knots = 3) were specified as predictors.

For phase 3, the relationship between pass rate and capture success was analysed using a generalised linear model (GLM), specifying a binomial error structure with a logit link function. Capture success of pregnant/lactating or juvenile barbastelle bats (*i.e.,* bats potentially suitable for radio-tracking to a maternity roost) was defined as the response variable, and the maximum bat passes recorded in that woodland during the citizen science surveys was defined as the predictor. ROC analysis was subsequently conducted, in the same manner as previously described, to determine the optimal cut-off threshold for bat activity (pass rate) likely to yield successful capture of barbastelle bats.

### Ethical approval

The work was approved by the Animal Welfare and Ethical Review Committee of the School of Life Sciences, University of Sussex (Ethical Application Ref: ARG-25). It was also part of work conducted under UK Home Office Licence number PIL P57B69020 and Natural England Licence no. 2022-61108-SCI-SCI and preceding licences.

## RESULTS

### Phase 1: Acoustic methodology development

We monitored bat activity for 470 detector-nights across 13 woodlands during the summer of 2019 (Table 1). Throughout the study, 1,549 barbastelle passes were recorded within the first hour of sunset, occurring at a mean time of 35 min (SD = 11.6) following sunset.

Of the 13 woodlands surveyed, historical evidence identified six sites as supporting colonies, whilst four sites were classified as having no colonies present. For the remaining three sites, we were unable to determine whether they were occupied or unoccupied by colonies based on our analysis of temporal activity patterns, and therefore these sites were classified as 'unknown' in our analysis. Among the sites without colonies, three exhibited minimal barbastelle activity throughout the entire survey period (range: 8–15 passes), with no clear peaks in activity. The remaining site recorded a substantial number of barbastelle

**Table 1 Summary of *B. barbastellus* passes recorded within the first hour of sunset at each site.** 'Max passes' represents the maximum number of passes recorded across all detectors during the survey period. 'Mean passes' represents the mean number of passes recorded across all detector-nights. All results presented are based off an analysis of the first hour after sunset.

| Site identity | Number of detectors (% locations species detected) | Detector-nights | Surveyed area (ha) | Mean passes | Max passes | Mean time of pass after sunset (mins) (SE) |
|---|---|---|---|---|---|---|
| *Colony present* | | | | | | |
| 1 | 10 (70) | 60 | 53 | 1.8 | 17 | 38.0 (0.91) |
| 2 | 15 (87) | 42 | 122 | 8.7 | 70 | 39.0 (0.66) |
| 3 | 14 (93) | 39 | 81 | 9.0 | 63 | 38.0 (0.55) |
| 4 | 13 (85) | 36 | 53 | 6.6 | 42 | 37.3 (0.65) |
| 5 | 12 (75) | 57 | 54 | 1.1 | 7 | 35.4 (0.84) |
| 6 | 14 (93) | 39 | 78 | 7.3 | 40 | 26.5 (0.52) |
| *Colony absent* | | | | | | |
| 7 | 14 (7) | 42 | 37 | 0.1 | 2 | 21.7 (0.21) |
| 8 | 10 (20) | 30 | 35 | 0.1 | 2 | 36.4 (3.73) |
| 9 | 14 (29) | 40 | 114 | 0.2 | 3 | 47.8 (2.47) |
| 10 | 6 (50) | 18 | 14 | 0.3 | 2 | 34.6 (3.92) |
| *Unknown* | | | | | | |
| 11 | 9 (34) | 36 | 24 | 0.7 | 6 | 28.9 (2.53) |
| 12 | 5 (60) | 16 | 23 | 0.8 | 3 | 34.6 (1.97) |
| 13 | 10 (50) | 30 | 75 | 3.0 | 18 | 31.2 (0.98) |

passes ($n = 180$), but very few within the first hour across the survey period ($n = 9$); corresponding with findings from a recent Environmental Impact Assessment which concluded that the site was used by barbastelles for foraging but not breeding (*Highways England, 2018*).

The number of bat passes within an hour of sunset strongly predicted the probability of colony presence or absence (GLM: OR $= 3.28$, 95% CI [1.24–8.65], $z_{1,33} = 2.40$, $p = 0.018$). Follow-up ROC analysis (AUC $= 0.966$, 95% CI [0.898–1]) identified that a cut-off point of 3.45 barbastelle passes within an hour of sunset optimises the balance between true-positive (sensitivity) and false-positive rates (1 - specificity) in predicting colony presence (Fig. 2). Using this value, 95.7% of survey nights in woodlands with known colonies, and 100% of survey nights where a colony was absent from the woodland, were correctly predicted. To accommodate practical considerations, this threshold is rounded up to four passes, as having a fraction of a pass is not possible. Subsequently, two woodlands that were previously unknown to contain colonies were identified as likely to support colonies (passes < 1 hour of sunset; site 11 = 6, site 13 = 18).

There was a significant non-linear relationship between increasing detector density and the probability of woodlands being correctly classed as positive for barbastelle roost occupancy (GAMM, Figs. S2 and S3). The effect of detector density on woodland classification was highly significant for each woodland ($p < 0.001$), with the survey effort required to reliably detect a colony varying depending on the woodland surveyed. Overall, there was a 90% chance of a woodland being correctly identified as having a colony with
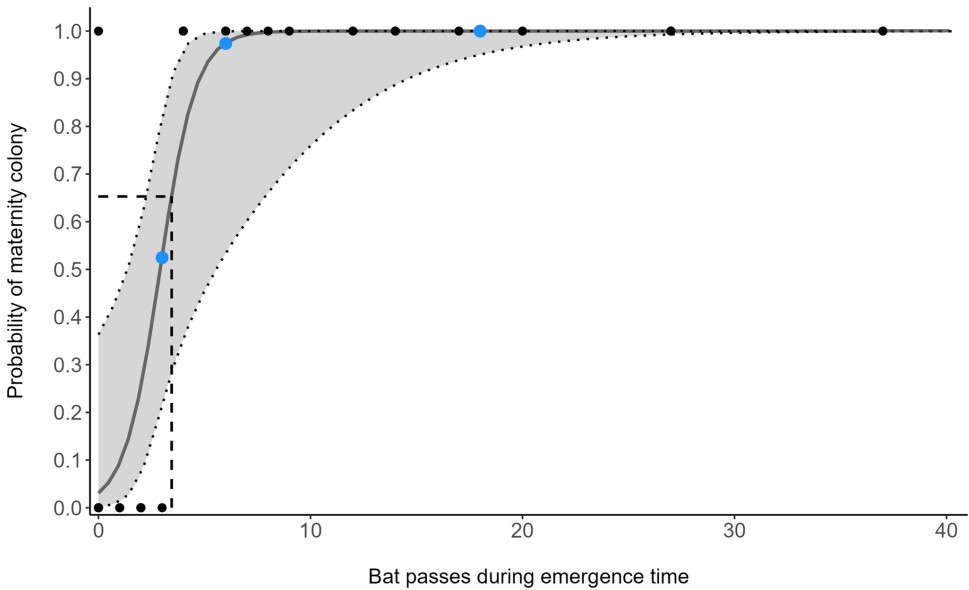

**Figure 2 Model predictions from a generalised linear model (GLM), showing the change of maternity colony probability with increasing *B. barbastellus* passes.** Points represent the number of barbastelle passes (per night) within the first hour after sunset in woodlands of known (black) and unknown (blue) colony status. Dotted lines/grey area represent the 95% confidence intervals. The dashed line represents the optimal cut-off threshold based on the Receiver Operating Characteristic (ROC) curve analyses.

a density of 0.16 detectors ha$^{-1}$ (Fig. 3). The adjusted-R$^2$ value for the model was 0.936. When only a single detector within a woodland was selected *via* our random subsampling approach, we found that there was just a 32% chance of detecting a colony based on the average survey effort. Survey duration had minimal effect on the ability to detect a colony, based on a survey sensitivity of 0.9 (Fig. 4). Each site showed similar detector densities needed to detect colonies, regardless of survey duration (GAM prediction ranges: site 1 = 0.106 − 0.107 detectors ha$^{-1}$, site 5 = 0.175 − 0.177 detectors ha$^{-1}$).

## Phase 2: Application of methodology

Based on the results of phase 1, volunteers undertook surveys using a minimum detector density of 0.16 detectors ha$^{-1}$ for a duration of three consecutive nights. Whilst the results of phase 1 indicate that a shorter survey duration may be sufficient to detect a colony, we chose three nights to account for the statistical uncertainty of our results due to a limited sample size. Surveys were completed at 509 different locations across 77 woodlands (three woodlands could not be surveyed due to time constraints). This comprised 1826 complete nights of recording, which resulted in the collection of 1,666 barbastelle recordings within the first hour after sunset. Barbastelles were recorded within one hour of sunset in 40 woodlands (52%), of which 26 (34%) were predicted to support colonies based on the optimal cut-off threshold identified in phase 1 (Fig. 5).

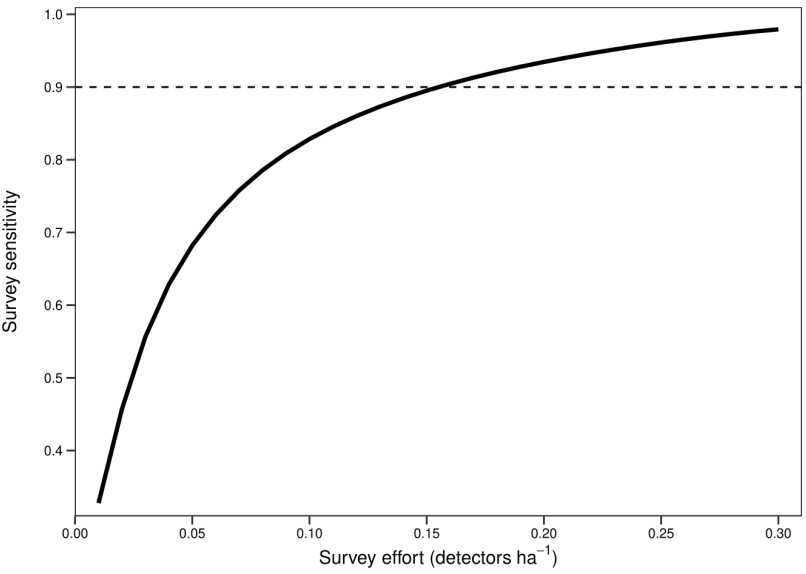

**Figure 3** **Overall survey effort required to detect colonies of *B. barbastellus* based on pooled data across sites.** Graph shows averaged predictions from a generalised additive mixed model (GAMM). Survey sensitivity represents the probability of correctly identifying a woodland with a colony, calculated for various detector densities. It is estimated as the proportion of 1,000 stochastic simulations in which at least one static detector in a woodland exceeds the optimal cut-off threshold of 3.45 bat passes (within one hour of sunset). Dashed line indicates the desired survey sensitivity of 0.90.

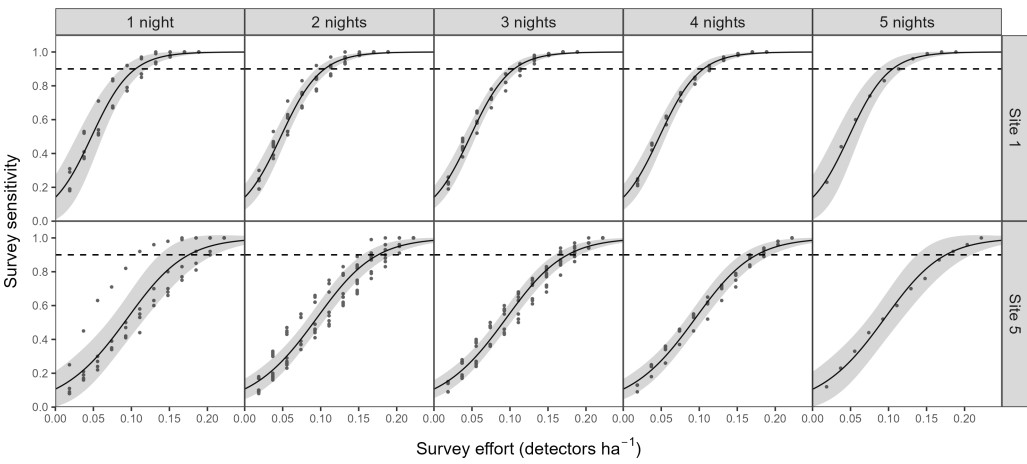

**Figure 4** **The impact of survey duration on the survey effort required to detect colonies of *B. barbastellus*.** Graph shows the predictions (solid line) from a generalised additive model (GAM), with 95% confidence intervals (grey area). Survey sensitivity represents the probability of correctly identifying a woodland with a colony, calculated for various detector densities. It is estimated as the proportion of 1,000 stochastic simulations in which at least one static detector across selected nights exceeds the optimal cut-off threshold of 3.45 bat passes (within one hour of sunset). Dashed lines indicate the desired survey sensitivity of 0.90.

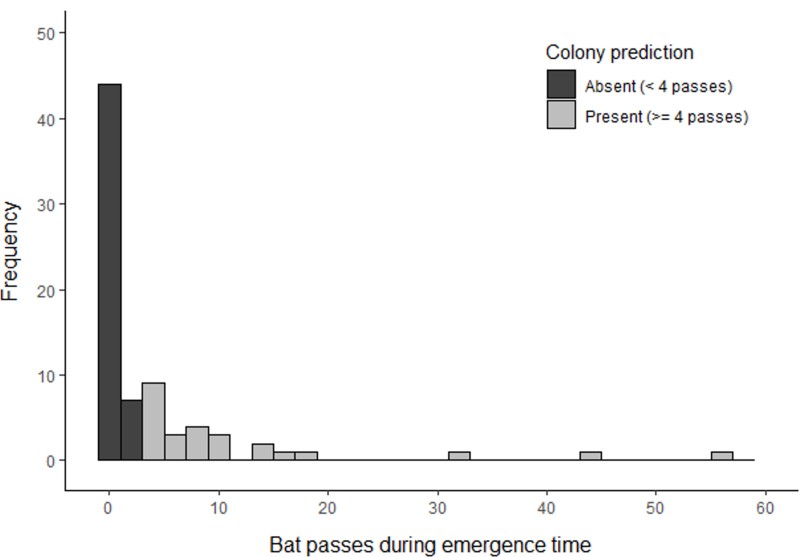

**Figure 5 Summary of *B. barbastellus* activity recorded across all sites surveyed by citizen scientists in 2021.** Data represents the highest number of barbastelle passes recorded across all detectors deployed within a woodland during the emergence period ($< 1$ hour after sunset).

## Phase 3: Validation by radio-tracking

Of the 26 woodlands predicted to support colonies, trapping surveys were conducted at 13 and barbastelles captured at 10. Trapping was also conducted at four sites that were predicted to have no colonies, and barbastelles were captured at one of these (a single adult male). Radio-tracking of 11 adult females and two juveniles in eight woodlands (those with bats suitable for tagging) led to the identification of five previously unidentified maternity colonies. Multiple roost trees were identified in two woodlands, though in both cases these were assumed to be part of the same colony due to observed roost-switching and proximity. All colonies were located within the woodlands where the bats were captured, and roost counts ranged from 14 to 52 bats (Table 2; $\bar{X} = 30.60$, SD = 15.55). Out of the seven tree roosts identified, four were found under exfoliating bark of standing dead trees. A further two roosts were identified within live trees - one within a shearing crack and the other in a crevice formed by a tree limb breaking off. The feature used for the remaining roost could not be identified as the three radio-tagged bats switched to a second roost between nights, and a further emergence survey could not be conducted due to time constraints. Of the woodlands predicted to have barbastelle colonies, barbastelles suitable for tagging (adult females or juveniles) were caught in 69% of sites, and the colony's roosting location was located in 38%.

The number of bat passes recorded within an hour of sunset was a strong predictor for the successful capture of bats associated with maternity colonies (adult females or juveniles) (GLM: OR = 1.24, 95% CI [1.11–1.44], $z_{1,71} = 3.35$, $p < 0.001$). The ROC analysis, based on the radio-tracking data, indicated that there would be a 33% chance of capturing barbastelles potentially suitable for radio-tracking back to colonies if three bat

**Table 2  Results of *B. barbastellus* radio-tracking surveys carried out in summer 2022.** 'Max passes' represents the maximum number of passes recorded (within an hour of sunset) across all detectors during the acoustic surveys conducted in 2021. 'Colony prediction' represents predictions based on the ROC analysis cut-off threshold. For all sites where barbastelles were caught but not tagged, the individuals were adult males and were considered unlikely to yield identification of a maternity colony and so were not radio-tracked. Adult females were prioritised for radio-tracking over juvenile bats.

| Site identity | Max passes | Colony prediction | Adult females | Adult males | Juvenile females | Juvenile males | Radio-tracking | | |
|---|---|---|---|---|---|---|---|---|---|
| | | | | | | | Tagged bats | Colony located | Emergence count |
| *Wiltshire* | | | | | | | | | |
| 1 | 14 | Present | 1 | 0 | 0 | 0 | 1 | Yes | 21 |
| 2 | 6 | Present | 2 | 2 | 0 | 0 | 2 | No | – |
| 3 | 0 | Absent | 0 | 1 | 0 | 0 | 0 | No | – |
| *South Midlands* | | | | | | | | | |
| 4 | 57 | Present | 2 | 2 | 3 | 3 | 2 | Yes | 52 |
| 5 | 44 | Present | 4 | 2 | 0 | 0 | 2 | Yes | 42 |
| 6 | 0 | Absent | 0 | 0 | 0 | 0 | 0 | No | – |
| *West Sussex* | | | | | | | | | |
| 7 | 10 | Present | 0 | 0 | 1 | 0 | 1 | Yes | 22 |
| 8 | 19 | Present | 0 | 2 | 0 | 0 | 0 | No | – |
| 9 | 6 | Present | 0 | 1 | 0 | 0 | 0 | No | – |
| 10 | 10 | Present | 0 | 0 | 0 | 0 | 0 | No | – |
| 11 | 0 | Absent | 0 | 0 | 0 | 0 | 0 | No | – |
| *Herefordshire* | | | | | | | | | |
| 12 | 32 | Present | 4 | 0 | 0 | 0 | 3 | Yes | 14 |
| 13 | 16 | Present | 1 | 1 | 0 | 0 | 1 | No | – |
| 14 | 4 | Present | 0 | 0 | 0 | 1 | 1 | No | – |
| 15 | 14 | Present | 0 | 0 | 0 | 0 | 0 | No | – |
| 16 | 4 | Present | 0 | 0 | 0 | 0 | 0 | No | – |
| 17 | 0 | Absent | 0 | 0 | 0 | 0 | 0 | No | – |

passes were recorded within the first hour after sunset at a detector (AUC = 0.856, 95% CI [0.754–0.9581]). When 16 or more bat passes were recorded then the probability of capture success increased to 90%.

# DISCUSSION

This project demonstrates that standardised acoustic surveys can be used to identify woodlands with a high probability of containing maternity colonies of rare bats, based on periods of high activity. Traditionally, acoustic surveys are used to identify bat foraging habitats and commuting routes, but are not typically used in a standardised way to identify roost sites. British guidance currently recommends the use of trapping and radio-tracking surveys for locating bat roosts in woodland habitat, given the low efficiency of conducting large-scale tree roost assessments (*Collins, 2016*). However, this is time consuming and requires highly trained specialists that have the relevant licences. We have shown that a triage approach can be used, where acoustic citizen science surveys are used to identify woodlands likely to contain the species. This level of survey on its own would be suitable

to fill key data gaps (such as monitoring trends in woodland occupancy rates, or to identify woodlands where management should account for the high probability of barbastelle colony presence); or it could be followed up with radio-tracking to provide more detailed evidence on the precise location and size of maternity colonies.

We found that a minimum density of 0.16 detectors per hectare is necessary to detect a barbastelle colony and maintain a survey sensitivity of $\geq$ 90%. Current recommendations for bat activity surveys *via* passive acoustic monitoring suggest surveying at least three locations per transect at sites with highly suitable habitat (*Collins, 2016*). Alternatively, in the case of wind turbines, it is suggested to use up to one detector per turbine depending on the size of the development (*NatureScot et al., 2021*). Whilst these recommendations are suitable for many surveys (*e.g.*, species presence, diversity, and a comparison of relative activity between sites) and have not been developed with the specific purpose of colony detection, they are proposed as a method to assess the potential impact of developments on bats. Our findings indicate that if we deployed a single detector at each site it would have resulted in only a 32% chance of detecting the hotspots of activity associated with a colony. This is not particularly surprising given that the former recommendations were developed in relation to transect-line (one-dimensional) and point surveys. Yet, many developments and conservation assessments apply to two-dimensional areas, which are often of considerable size. Therefore, single detectors may not always provide sufficient opportunities to detect activity hotspots, especially if individuals favour some areas over others. Consequently, temporal replication (*i.e.*, longer survey duration) cannot be substituted for spatial replication. *Kubista & Bruckner (2017)* likewise demonstrated the limitation of deploying just a single detector per site, with only 17% of their 157 study sites in Austria registering the same species diversity across three batcorders placed in close proximity (ca. 10 m apart). In addition, they found that recording performance was significantly affected by increasing vegetation density and species with a short maximum call range *e.g.*, *Rhinolophus hipposideros*. At one of our sites (Site 2, Table 1), four detectors out of the 15 deployed did not record any passes within an hour of sunset, despite the woodland as a whole recording high levels of barbastelle activity and a colony being present in the woodland. Given that many studies deploy only a single acoustic monitoring device at a site (*Sugai et al., 2019*), this has profound implications for ecological surveys that are intended to inform conservation action.

Radio-tracking surveys confirmed the presence of colonies in five out of the 13 woodlands predicted to have a colony. In addition, we captured lactating/post-lactating females and juveniles at another three sites but were unable to locate colonies through radio-tracking, though their presence suggests that colonies may have been nearby. In contrast, we caught only one individual across the four woodlands that were predicted not to contain colonies, and this was an adult male unlikely to have been associated with a breeding colony. We deliberately deployed a standardised amount of trapping effort that took into account both practical and logistical feasibility, and might reasonably be used in a citizen science survey by trained bat workers. However, there are two explanations as to why barbastelles may not have been caught in woodlands predicted to have them. Either survey effort was insufficient, or the predictions themselves were incorrect (*i.e.,* optimal cut-off threshold

was too low). It is not possible to distinguish between these two explanations, and therefore we regard it important to consider both possibilities. Due to this uncertainty, we took a precautionary approach and did not conduct further ROC analysis that incorporated this ground-truthed data, as any refinement (*i.e.,* a higher cut-off threshold) would likely result in an increase in colonies going undetected. Alternatively, as the trapping was conducted a year after the acoustic surveys for logistical reasons, it may have given the opportunity for colonies to move to other locations in the interim.

There is a trade-off between survey sensitivity and survey effort. ROC analysis allows us to optimise the cut-off point by balancing the need for a high true-positive rate whilst also minimising false-positive results. This ensures trapping effort is focused in areas of high priority. Typically, it is more serious to make a false-positive claim (Type I error) than a false-negative one (Type II error), and therefore it is conventional to use an 80% threshold in power analyses (*Cohen, 1992*). However, where it is critically important that a roost is not overlooked, such as a major infrastructure project, we recommend using a 90% threshold (*i.e.,* one detector per 6.25 ha) or higher, as adopted within this study. On the other hand, where there is no direct threats to roost sites, such as when the aim is to understand the conservation status of a species in a region or provide more general management guidance, then a lower survey sensitivity could be considered. For example, using a survey sensitivity of 80% (*i.e.,* one detector per 11.11 ha) would considerably reduce the number of detectors needed and the time requirements for deployment, as well as the amount of data analysis. Alternatively, larger woodlands could be surveyed in sections, as was the case for several sites within this study.

The approach we present here could be widely deployed and adapted to establish survey designs in other geographical regions (*e.g.*, Continental Europe) or for other bat species. However, precise methodological parameters may need to be modified on a case-by-case basis owing to species-specific variation in detection probability (*Mackenzie & Kendall, 2002*), emergence time (*Jones & Rydell, 1994*; *Thomas & Jacobs, 2013*), call intensity (*Britzke, Gillam & Murray, 2013*), and average colony size (*Rueegger, Law & Goldingay, 2018*). Citizen science projects, such as The National Bat Monitoring Programme in Britain, have previously demonstrated the value that survey data collected by trained volunteers can have for modelling bat population trends (*Barlow et al., 2015*).

On the basis of our results, we suggest a triage system for identifying woodland sites with a high probability of occupancy by barbastelle colonies, and the screening out of lower probability sites, based on acoustic survey data. We propose the following protocol:

(i) Calls should be collected using digital recording methods.

(ii) Detectors should be placed along trails and rides where possible. Approximate equal spacing should be maintained but opportunistic placement of detectors is acceptable.

(iii) To maintain a 90% probability of detecting colonies where they are present, detectors should be placed at a minimum density of 0.16 detectors ha$^{-1}$ for three nights. This could be reduced to 0.09 detectors ha$^{-1}$ if an 80% sensitivity is acceptable.

(iv) Microphones should be raised ∼1.5 m from the ground and orientated horizontally towards the path and away from vegetation.

(v) Recordings should be manually verified using a sound analysis programme.

(vi) To identify a woodland as likely to contain a colony, at least one detector must record four or more barbastelle passes within the first hour after sunset.

(vii) Where precise information is needed on roost location or size, acoustic surveys could be followed up with radio-tracking, with trap locations being guided by the locations of detectors with the most bat calls.

## CONCLUSIONS

Each traditional method for identifying colonies of barbastelles has inherent advantages and disadvantages. This study demonstrates that a relatively simple fieldwork methodology can enable the 'screening' of woodlands that have the potential for supporting colonies. Acoustic surveys can offer new insights on colony activity from those obtained from trapping and radio-tracking studies, and we show that it has the potential to facilitate these surveys and contribute to more efficient and focused radio-tracking efforts. By being able to identify, both rapidly and cost-efficiently, the woodlands with high colony potential, we can contribute to the protection of woodland habitats and the species that use them. We urge future studies to develop the application of acoustic surveys to monitor colonies, as well as extending this methodology to other rare bat species.

## ACKNOWLEDGEMENTS

We would like to thank the volunteers of Vincent Wildlife Trust who contributed to the collection of acoustic data for this project, as well as Laura Lawrance-Owen at Vincent Wildlife Trust who helped with volunteer recruitment and organisation. Thanks also to Zoe Benefer and Lucas Cox for assisting with fieldwork. We are also grateful for the support and assistance from the Sussex Bat Group, Wiltshire Bat Group, Bedfordshire Bat Group, North Bucks Bat Group, Herefordshire Mammal Group, and Nene Valley Bat Group. Thank you to Simon Smart, Bob Cornes, Steven Allen, and Emily Dickens for their assistance with land access and bat surveys. Lastly, we would like to thank Gareth Harris for his continued support throughout this project, lending both his knowledge and fieldwork expertise at every stage, as well as helping with land access.

### Funding

This work was conducted as part of a PhD funded by the University of Sussex and Vincent Wildlife Trust. Former and current staff of Vincent Wildlife Trust provided support in data collection. Kieran D. O'Malley is a PhD student joint supervised by Prof Fiona Mathews (main supervisor) and Dr. Henry Schofield (secondary supervisor), both of whom contributed to study design and/or data collection.

### Grant Disclosures

The following grant information was disclosed by the authors:
The University of Sussex and Vincent Wildlife Trust.

## Competing Interests

Kieran D. O'Malley is a PhD student joint funded by the University of Sussex and Vincent Wildlife Trust. Henry Schofield and Tom Kitching are former employees of Vincent Wildlife Trust. Patrick G.R. Wright, Daniel Hargreaves, and Marina Bollo Palacios are employed by Vincent Wildlife Trust. Fiona Mathews is employed by the University of Sussex.

## Author Contributions

- Kieran D. O'Malley conceived and designed the experiments, performed the experiments, analyzed the data, prepared figures and/or tables, authored or reviewed drafts of the article, and approved the final draft.
- Henry Schofield conceived and designed the experiments, authored or reviewed drafts of the article, and approved the final draft.
- Patrick G.R. Wright performed the experiments, authored or reviewed drafts of the article, and approved the final draft.
- Daniel Hargreaves performed the experiments, authored or reviewed drafts of the article, and approved the final draft.
- Tom Kitching performed the experiments, authored or reviewed drafts of the article, and approved the final draft.
- Marina Bollo Palacios performed the experiments, authored or reviewed drafts of the article, and approved the final draft.
- Fiona Mathews conceived and designed the experiments, performed the experiments, authored or reviewed drafts of the article, and approved the final draft.

## Ethics

The following information was supplied relating to ethical approvals (i.e., approving body and any reference numbers):

The work was approved by the Animal Welfare and Ethical Review Committee of the School of Life Sciences, University of Sussex (Ethical Application Ref: ARG-25). It was also part of work conducted under UK Home Office Licence number PIL P57B69020 and Natural England Licence no. 2022-61108-SCI-SCI and preceding licences.

## Data Availability

The raw data is available in the Supplementary Files.

## Supplemental Information

Supplemental information for this article can be found online at http://dx.doi.org/10.7717/peerj.15951#supplemental-information.

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
