# Peer review of "An acoustic-based method for locating maternity colonies of rare woodland bats"

_PeerJ, doi:10.7717/peerj.15951_

## Round 0.1 · original submission · Minor Revisions

Please re-submit the paper after incorporating the all comments of reviewers. Moreover, please improve the language of the paper and explanations about the sample.

Reviewer 1 ·

Basic reporting

The manuscript is very well written, structured and presented. The topic is interesting and applicable to improving and standardising survey methods for bats. The statistical analysis is appropriate. I believe the manuscript should be published following the inclusion of the recommended changes and further clarification input described in the following sections.

Experimental design

The research is orginal and in scope of the journal. The research question is well defined and relevant to research and industry methodologies.

Line 118. Does ‘previous records of known colonies (n = 6)’ refer to barbastelles? If so, add this for clarity.

Line 168. I do not see how figure S1 relates to the sentence.

Line 218. Why did you select three nights of recordings? Is this important for your result? It seems important given a large proportion of your discussion is dedicated to pushing that more detectors are superior to more time. Please expand on this.

Carr et al. 2022 (https://doi.org/10.1016/j.foreco.2022.120680) found that three nights recordings in these types of small fragmented woodlands would give 0.85 probability of recording all bats present. This indicates species such as barbastelles and plecotus are not always picked up at night three (although single detectors were used, the size of the woodlands would put the number of detectors in line with your recommended one per 6.25 ha).

Line 267. Please state how many detector failures.

Line 227. Why did you only catch and track at four low probability and 13 high probability woodlands. This needs justifying as it feeds into the GLM analysis and is unbalanced without any explanation as to why.

Line 274. Is the term ‘error’ needed. Is it not fine just to state binomial?

Line 325 and figure 2. The use of ROC with the binomial GLM seems very logical and elegant. However, after reading the text and related figure I am unsure how you came to four passes as a threshold. It seems this number of passes is ‘tipping the balance of probability’. If I follow this correctly having four passes brings the probability just above 50%. Being just over 50/50 does not seem like a good threshold. Looking at the figure if the GLM without the ROC analysis it seems 5 passes (0.95) or even 6 passes (to account for confidence intervals) is more suitable. Please provide more information on the ROC analysis that covers the above concern. Specifically, please explain how you get an AUC of 0.966 with four passes and why this is considered a threshold.

Did you check the model output for normality of residuals? If so, please state this. If not, please explain what steps were taken to check model output.

Validity of the findings

A concern of the results and limitation of the conclusion is that of the 13 woodlands you failed to find barbastelles in five of them. What does this mean for the validity of the model and the ROC threshold used? It also needs some explanation as to why only four sites predicted not to have colonies were surveyed.

You ground truthed the model output so you could include the results of this using positives true/false, and negative true/false results (ROC). I suggest you complete further analysis and add the results – or justify why this is not needed.

Line 412 – 417. The explanation of why you did not catch barbastelles in some sites predicted to contain colonies is poor. Firstly, why did you use a standardised amount of survey effort that could be reasonably used by trained bat workers? This research is looking to prove a concept and so you could have prioritised maximising your survey effort. Secondly, stating that if you would have undertaken more survey effort you would have likely caught barbastelles in another five woodlands is speculation. Line 419 is more realistic ‘some woodlands were incorrectly predicted to contain colonies’. I recommend that lines 412 – 417 are removed and an acknowledgment that the threshold used may be too low is used as a replacement.

Additional comments

Title. Is ‘triage approach’ the best wording? Triage is to identify which problems or tasks should be dealt with first. I suggest a more direct title: An acoustic-based method for locating maternity colonies of rare woodland bats.

Line 58. The term ‘richly structured’ is not informative. For barbastelles there are described characteristics that influence their presence in woodlands (see Carr et al, 2022 https://doi.org/10.1016/j.foreco.2019.117682, Russo et al 2017 https://doi.org/10.1002/ece3.3111. and many others. I would suggest being more descriptive here.

Line 59. Saying our understanding of barbastelle bat colonies is currently limited is unfair. There has been a lot of focus throughout Europe on barbastelles since 2005. The statement should follow that although there are still knowledge gaps our understanding of barbastelle colonies in woodlands has improved.

Line 86 to 98. Here you are making the point that deploying more detectors over a shorter period maybe a more efficient approach than less (or one detector) over a longer time period. Did Fischer et al. (2009) actually demonstrated this? I would recommend reworking this paragraph to be clearer and include the important point that optimising methodology depends on the objective. The 70% of studies that deployed just a single recorder per site likely all had the objective to sample bats and compared species number and relative activity between sites. This is a suitable approach for that objective, provided they were deployed for a long enough period. Your objective is to find ‘hotspots’ so it is obvious that a spatial element is needed.

Line 91. Add detail for ‘spatially biased’.

Line 111. What about temporal requirements for survey effort?

Line 168. Should ‘Fig S1’ be on line 167 after the first sentence?

Line 197. This information is useful for understanding woodland requirements of barbastelles. It seems buried. I would suggest making this objective more apparent in the introduction.

Line 244. Replace ‘2’ with ‘two’

Line 245. Replace ‘bats’ with ‘barbastelles’

Line 338. It is not clear when you state ‘mean sensitivity when deploying a single detector’ where this value comes from.

Line 345 – 350. It is difficult to follow what information is being presented. I had to read the text several times to work out that ‘predicted to support’ and ‘likely to contain’ are different. This makes perfect sense but the way the text is presented reduces the clarity. I suggest a rewrite of these sentences.

Line 344. Remove ‘across six counties’. It is not needed and results in confusion.

Line 358. Please expand on the types of roosts found. This is interesting information as the species has been observed selecting roosts under exfoliating bark despite having available crevice roosts in broadleaved trees. As a reader I would like more detail. It is not fully clear if the roosts were tree cavities or exfoliating bark.

Line 375. Replace ‘rarely used’ with ‘used as standard’ (or something similar). There are many occasions in research and ecological consultancy that acoustic surveys (back tracking or general deployment of detectors to consider likely use) are used to find maternity colonies. I agree this is not standardised though.

Line 378. Add the fact licences are required.

Line 391 -393. You did not look at this specifically and so it seems out of place. It is not correct to say rare species when your data looked at only one species (that itself is questionable if it can be considered rare within its geographic range). I recommend this statement is removed or at least reworded. You then write in the next sentence that you found a single detector has only a 32% chance of detecting a colony. This is written to relate to the previous sentence but those surveys are interested in presence and you present results that are interested in colonies – these are different!

Line 398 – 401. You use an example of one your woodlands to critique the use of single detectors per site but the example woodland you use had four detectors. This does not make any sense.

Line 403-406. Is the interpretation of Gorresen et al. (2008) correct? I am not convinced by the argument that spatial cover should be prioritised over temporal cover. Time does result in more species recorded. Temporal cover also allows changes in presence over time and through the seasons. I think a more balanced view is needed than simply pushing the argument that more detectors are better than less. In an ideal world more detectors and more time is optimal.

Line 410. The presence of lactating barbastelles in your study landscape does not suggest a colony is nearby – lactating barbastelles in fragmented landscapes are documented making regular nightly commutes of 8 km or more. The presence of juveniles similarly does not suggest a colony is nearby. Why did you not radio track these lactating/post lactating and juvenile bats?

Line 445 – 447. No need to make this point.

Line 462. Change ‘site’ to ‘woodland’.

Line 471. Remove ‘rapid’. Your proposed method is far from rapid.

Reviewer 2 ·

Basic reporting

See section 2 for summary review of the ms, and other comments

End of intro L110-113: there seem to be other questions addressed in the methods that are not outlined here (or perhaps they are steps to answering these questions, but this needs to be clearer). For instance, L194 (phase 2) explains that the site selection enables an investigation of whether patch size influenced the probability of occupancy (and repeated in 197-198), but this isn’t a question posed in the introduction, there is no analysis of this, and it is not presented in the results. If this means for future analyses not presented in this paper then this just needs some clarification.

Experimental design

Summary
This is a hugely valuable paper, based on rigorous fieldwork with an impressive sample size and some very practical conservation recommendations. Congratulations to all involved. It is largely very well written and easy to follow. Most of my comments are very minor but three areas which need more attention are:
1) More information on site selection for phase 1 is needed. In L118-199 it is not clear whether the aim was to include a mix of sites with known presence and presumed absence, or only sites where the species was thought to be present. Numbers are needed here. L153 suggests that the species was not thought to be present at some sites – what was the aim here, a 50:50 split, or was it more opportunistic based on sites with appropriate information?
2) The statistical analysis section needs to be much clearer on which question (and which phase) is being addressed by the various analyses. For example, in L303 outlining how the relationship between pass rate and capture success, it is not clear to me which of the two questions outlined at the end of the introduction is addressed here – we either need a clearer picture of the steps taken to address each of these wider questions, or these additional questions need to be outlined more clearly in the introduction.
3) I felt there was a degree of over-reach on recommendations, with more recognition needed that surveys are carried out for all sorts of reasons, and not just those of the current study, so that there are other considerations for survey design.

Minor comments
Introduction
L35-37 – clarify that the radiotracking surveys in a subset of woodlands were those sites where acoustics predicted a high probability of a colony being present.
L40-41 – since conservation status will no doubt be dependent on follow up specialist surveys it would probably be better to switch around the order here, so it’s about prioritising sites for future specialist surveys and conservation action?
L47 – not just presence, but also population size.
L48 – including whether a species is detected is also a product of survey effort (this may be implied in the current text but could be clearer).
L86-89 – Acoustic surveys are used to address a wide range of questions, so the design is also influenced by these and may also involve a consideration of sample size. If this sentence is specifically referring to surveys to determine species presence at particular sites (rather than say, factors influencing occupancy across a large number of sites), then some clarification would be useful here.

Methods
L121-123 – is this a general climatic statement (rather than for 2019)? the source of this information be useful.
L144 – a definition of bat pass is needed here
L148 – “close to dawn may also be indicative…”? Given the preceding sentence I think the word also is needed here.
L153 – please clarify how old the historical evidence was for this
L157-158 – were these sites those where colonies were not thought to be present? I’m not clear what the purpose of including this sentence in the methods section is?
L161 – are these 3 sites additional to the original 13? I found this paragraph quite confusing in general as results for some (but not all) sites are described. It might be easier to follow if these results were presented in the results section, not methods.
L167-168 – I don’t feel that Fig S1 is necessary since it just visualises a straightforward division of woodland size by detectors. It also doesn’t say anything about how the heterogeneity increases with the size of a site (but where this fig is cited suggests that it does).
L184 – were any of the sites used in Phase 1 were included here?
L194-200 – this felt like quite a convoluted explanation and could be more concise. Could you just say that there were four categories of woodland and then provide some justification. Also 194-195 is a bit of a mouthful. You could say ‘… defined the minimum size…’ ??
L228-229 – what was the rationale for 1 low probability vs 13 high probability sites? Give the total number of sites where trapping is conducted (presumably 14 x 4 = 56).
L271 – see earlier comment about this… the Phase 1 methods don’t say anything about sites where absence was known (or suspected).

Results
L330-331 – were these sites subsequently confirmed to have colonies by trapping/radiotracking?
L335 – Fig S3 also demonstrates that survey effort required varies between sites (not just that there is a non linear relationship).
L346 – “… predicted based on the optimal cut-off threshold identified in phase 1”.

Discussion
L389-390 – detection of suitable habitat for foraging bats and detection of colonies are not the same thing so I don’t think this is a fair comparison to make.
L393 – the issue presented here is not about variability amongst detectors. This needs rephrasing.
L402 – I feel this is a broader statement that is justified as this is really dependent on the purpose of the survey. For the aim that this current study has then yes this seems reasonable, but there are many ecological studies which have different aims but may still be informing conservation. For example, for some questions a large number of sites may be needed which can necessitate reduced sampling intensity per site individual sites.

Fig. 2
This figure would benefit from larger points and it being easier to distinguish black from dark blue. Currently the sites with a probability of zero are cut in half so the y axis should be extended a little. I would also like to see which sites were felt to have colonies present or absent based on the historical information mentioned in the methods.

Validity of the findings

No comment

---

## Round 0.2 · accepted · Accept

The manuscript has been improved by incorporating the comments of reviewers. As per my reading, all minor comments have been incorporated appropriately and the manuscript is ready and accepted for publication.